# Nrf2 signaling links ER oxidative protein folding and calcium homeostasis in health and disease

Veronica Granatiero, Csaba Konrad, Kirsten Bredvik, Giovanni Manfredi, Hibiki Kawamata

**We report a signaling pathway linking two fundamental functions of the ER, oxidative protein folding, and intracellular calcium regulation. Cells sense ER oxidative protein folding through $H_2O_2$, which induces Nrf2 nuclear translocation. Nrf2 regulates the expression of GPx8, an ER glutathione peroxidase that modulates ER calcium levels. Because ER protein folding is dependent on calcium, this pathway functions as rheostat of ER calcium levels. Protein misfolding and calcium dysregulation contribute to the pathophysiology of many diseases, including amyotrophic lateral sclerosis, in which astrocytic calcium dysregulation participates in causing motor neuron death. In human-derived astrocytes harboring mutant SOD1 causative of familial amyotrophic lateral sclerosis, we show that impaired ER redox signaling decreases Nrf2 nuclear translocation, resulting in ER calcium overload and increased calcium-dependent cell secretion, leading to motor neuron death. Nrf2 activation in SOD1 mutant astrocytes with dimethyl fumarate restores calcium homeostasis and ameliorates motor neuron death. These results highlight a regulatory mechanism of intracellular calcium homeostasis by ER redox signaling and suggest that this mechanism could be a therapeutic target in SOD1 mutant astrocytes.**

## Introduction

The ER is the major intracellular calcium store, involved in a number of fundamental calcium signaling pathways. The mechanisms of ER calcium regulation are intimately connected with the redox state of the ER because the functions of ER calcium regulatory proteins are modulated by oxidative modifications (reviewed by Appenzeller-Herzog & Simmen (2016)). The redox state of the ER depends on oxidative protein folding mechanisms, whereby nascent proteins imported into the ER for secretion are folded into disulfide bond-containing secondary structures by the transfer of electrons from their thiol groups to ER oxidoreductases. Oxidoreductases are then reoxidized by transferring electrons to molecular oxygen. For every disulfide bond formed, one molecule of $H_2O_2$ is produced (Zito, 2015).

$H_2O_2$ produced by oxidative protein folding in the ER could contribute to up to 25% of reactive oxygen species (ROS) produced during protein synthesis (Tu & Weissman, 2004). Although the ER redox state is known to directly modulate ER calcium protein activity through cysteine modifications (Li & Camacho, 2004; Higo et al, 2005; Marino et al, 2015; Ushioda et al, 2016), the signaling role of ROS produced by oxidative protein folding and diffused outside of the ER (Appenzeller-Herzog et al, 2016) in regulating ER calcium stores remains to be elucidated.

We hypothesized that ER oxidative protein folding could initiate a ROS signaling pathway that involves both intra- and extra-ER components and regulates ER calcium homeostasis and signal transduction. We deemed that testing this hypothesis would be important for better understanding the link between ROS signaling and intracellular calcium regulation. We show that $H_2O_2$ produced by ER oxidative protein folding regulates nuclear factor E2-related factor 2 (Nrf2) signaling, which in turn modulates the expression of glutathione peroxidase 8 (GPx8), a protein involved in ER calcium regulation through the sarco/endoplasmic reticulum calcium ATPase (SERCA) (Yoboue et al, 2017). Thus, these findings identify an axis linking ER oxidative protein folding to calcium signaling through ROS and Nrf2.

Alterations of the oxidative protein folding-Nrf2-ER calcium axis could be involved in pathological conditions, such as neurodegenerative diseases, where dysregulation of ROS and calcium signaling are prominent features. We tested this hypothesis in a cell culture model of familial amyotrophic lateral sclerosis (ALS) with Cu/Zn superoxide dismutase (SOD1) mutation. We show that mutant SOD1 astrocytes, which are known to play an active pathogenic role (Ilieva et al, 2009), have an impairment in the oxidative protein folding-Nrf2-ER calcium axis. This finding is consistent with observations of Nrf2 alterations in animal models of SOD1 familial ALS (Mimoto et al, 2012; Mead et al, 2013). Moreover, we find that as a consequence of calcium dysregulation, mutant SOD1 human astrocytes have increased mitochondrial energy metabolism and ATP levels, resulting in enhanced calcium-dependent cell secretion and leading to motor neuron death in an astrocyte–neuron coculture system. Importantly, activation of Nrf2 signaling with dimethyl fumarate (DMF) in mutant SOD1 astrocytes effectively rescues these defects and attenuates the induction of motor neuron death.

---

Feil Family Brain and Mind Research Institute, Weill Cornell Medicine, New York, NY, USA

Correspondence: hik2004@med.cornell.edu

## Results

### H₂O₂ signaling links ER oxidative protein folding to ER calcium regulation through Nrf2-dependent GPx8 expression

To test the hypothesis that ER oxidative protein folding is functionally linked to ER calcium homeostasis through ROS, we treated HeLa cells with a selective Ero1 inhibitor (EN460), which was shown to block ER protein folding in cultured cells at 50 $\mu M$ (Blais et al, 2010). Upon EN460 treatment (50 $\mu M$ for 90 min), we observed a decrease in the fluorescence signal of the H₂O₂ sensor HyPer in the cytosol (Fig 1A), indicating decreased H₂O₂ levels consistent with impaired ER protein folding leading to decreased ROS production.

H₂O₂ signaling results in the release of Nrf2 from a complex with Kelch-like ECH-associated protein 1 (Keap1) and the ubiquitin ligase cullin 3, allowing for Nrf2 translocation to the nucleus, where it binds to the antioxidant response element (ARE) and activates the expression of its target genes (Tonelli et al, 2017). We hypothesized that Nrf2 target genes could be involved in ER calcium regulation. The ER resident glutathione peroxidase 8 (GPx8) was shown to regulate ER calcium by modulation of SERCA (Yoboue et al, 2017). Because glutathione peroxidases are known targets of Nrf2 (Tonelli et al, 2017), we

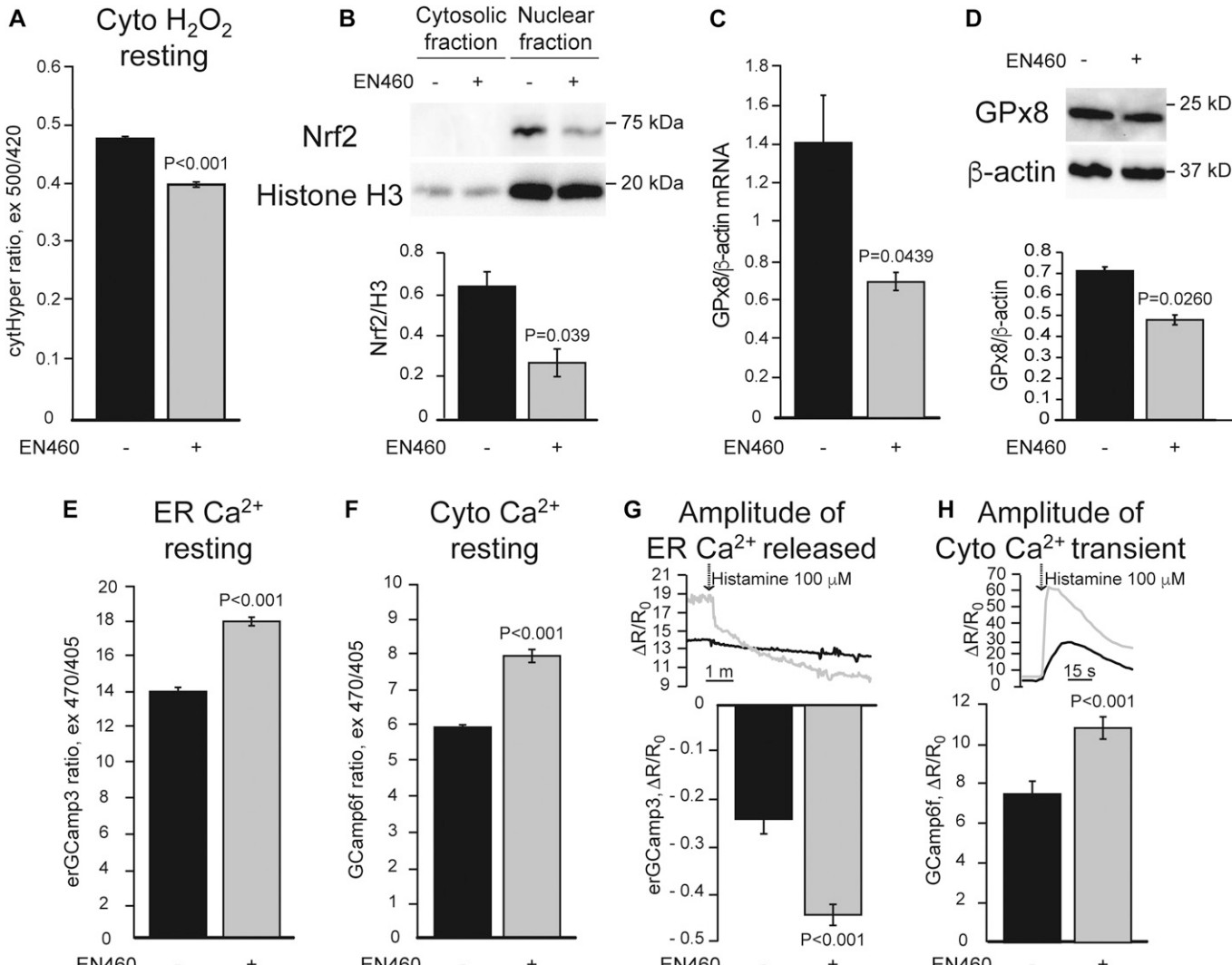

**Figure 1.  ER oxidative protein folding inhibition leads to deficient Nrf2 signaling and ER calcium dysregulation.**
HeLa cells were treated with 50 $\mu M$ Ero1 inhibitor, EN460 (+), or vehicle (−) for 90 min before the assays. All data are presented as mean ± SEM. **(A)** Basal cytosolic H₂O₂ content measured with HyPer (cytHyPer). n = 276, 305 cells from three sets of experiments. *P* value by Mann–Whitney test is indicated. **(B)** Representative Western blot of Nrf2 in the nuclear and cytosolic fractions, and quantification of nuclear Nrf2 normalized by the nuclear marker Histone H3 (H3). n = 3. *P* value by unpaired *t* test is indicated. **(C)** GPx8 mRNA levels normalized by β-actin. n = 3. *P* value by unpaired *t* test is indicated. **(D)** Representative Western blot and quantification of GPx8 protein levels normalized by β-actin. n = 3. *P* value by unpaired *t* test is indicated. **(E)** Resting ER calcium content measured by ER targeted GCaMP3 (erGCaMP3). n = 249, 284 cells from two sets of experiments. *P* value by Mann–Whitney test is indicated. **(F)** Resting cytosolic calcium levels measured with GCaMP6f. n = 260, 283 cells from two sets of experiments. *P* value by Mann–Whitney test is indicated. **(G)** ER calcium released with 100 $\mu M$ histamine, measured by the decline in erGCaMP3 fluorescence ratio. n = 33, 33 cells from two sets of experiments. *P* value by Mann–Whitney test is indicated. **(H)** Cytosolic calcium peak, measured by GCaMP6f, in response to ER calcium release induced by 100 $\mu M$ histamine. n = 50, 61 cells from two sets of experiments. *P* value by Mann–Whitney test is indicated.

assessed if EN460 affected Nrf2 nuclear translocation and expression of GPx8. In cells treated with EN460, Nrf2 levels in isolated nuclei, assessed as the amount of Nrf2 relative to histone H3, were decreased (Fig 1B). This was accompanied by a decrease in mRNA levels of known ARE-containing genes, heme oxygenase 1 (HO-1) and NAD(P)H quinone dehydrogenase 1 (NQO1) (Fig S1A and B). Importantly, GPx8 mRNA and protein levels (Fig 1C and D) were also reduced, demonstrating that the levels of GPx8 are regulated by ER protein folding activity.

Because GPx8 regulates ER calcium, to assess the effects of ER oxidative folding inhibition on calcium homeostasis, we assessed resting calcium levels and calcium dynamics in HeLa cells using genetically encoded calcium indicators. Ero1 inhibition led to increased resting ER and cytosolic calcium (Fig 1E and F) as well as increased ER calcium release in response to metabotropic stimulation with histamine (Fig 1G and H). Taken together, these results suggest that inhibition of ER oxidative protein folding reduces $H_2O_2$ signaling, which dampens Nrf2 translocation to the nucleus, leading to a decrease in GPx8 that results in increased ER calcium content and signaling.

To confirm the involvement of Nrf2 in ER calcium regulation through GPx8, we used a genetic approach. Nrf2 silencing in HeLa cells (Fig 2A) resulted in reduced levels of the protein in the nucleus

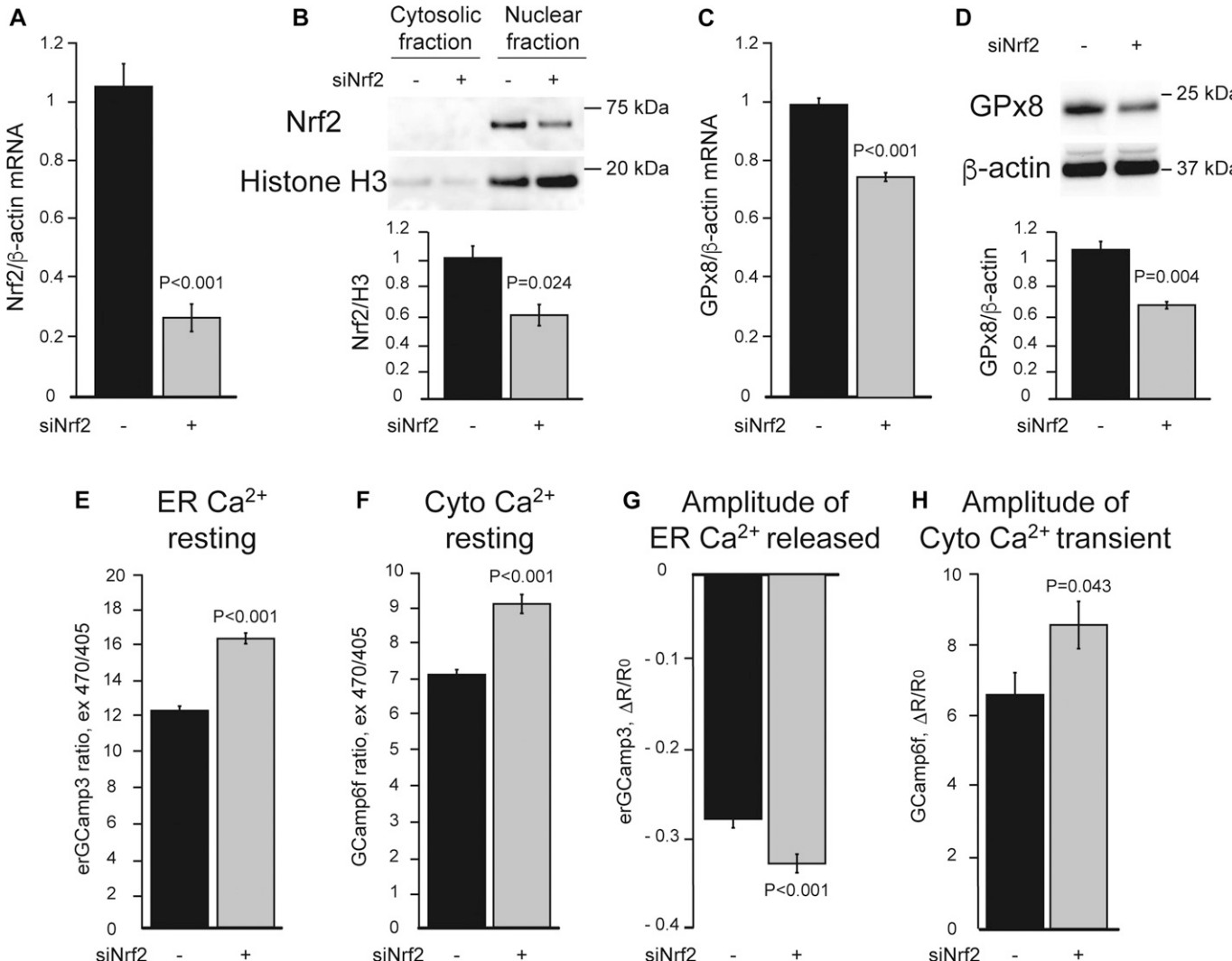

**Figure 2. Nrf2 silencing leads to ER calcium dysregulation.**
HeLa cells were transfected with Nrf2 siRNA (+) or with Renilla Luciferase siRNA as negative control (−) for 24 h before the assays. All data are presented as mean ± SEM. **(A)** Nrf2 mRNA levels, normalized by β-actin in cells with Nrf2 silencing. n = 3. *P* value by unpaired *t* test is indicated. **(B)** Representative Western blot of nuclear and cytosolic Nrf2 levels. Quantification of Nrf2 in the nucleus, normalized by Histone H3. n = 3. *P* value by unpaired *t* test is indicated. **(C)** GPx8 mRNA levels normalized by β-actin. n = 3. *P* value by unpaired *t* test is indicated. **(D)** Representative Western blot and quantification of GPx8 protein levels normalized by β-actin. n = 3. *P* value by unpaired *t* test is indicated. **(E)** Resting ER calcium content measured with erGCaMP3. n = 189, 223 cells from two sets of experiments. *P* value by Mann–Whitney test is indicated. **(F)** Resting cytosolic calcium levels measured by GCaMP6f. n = 272, 313 cells from two sets of experiments. *P* value by Mann–Whitney test is indicated. **(G)** ER calcium released upon 100 μM histamine stimulation, measured with erGCaMP3. n = 57, 50 cells from two sets of experiments. *P* value by unpaired *t* test is indicated. **(H)** Cytosolic calcium peak, measured by GCaMP6f, in response to ER calcium released induced with 100 μM histamine. n = 45, 46 cells from two sets of experiments. *P* value by Mann–Whitney test is indicated.

(Fig 2B) and decreased HO-1 and NQO1 expression (Fig S1C and D). Accordingly, GPx8 mRNA and protein levels were also reduced (Fig 2C and D), resulting in an increase in resting ER and cytosolic calcium levels as well as histamine-induced ER calcium release (Fig 2E–H).

Next, we tested the effects of increasing Nrf2 signaling using DMF, a well-characterized activator of Nrf2 (Brennan et al, 2017; Hayashi et al, 2017), with and without Ero1 inhibition. In EN460 treated HeLa cells, DMF completely prevented the decline in nuclear Nrf2 (Fig 3A), HO-1, and NQO1 (Fig S1E and F). Importantly, DMF also prevented the decrease in GPx8 mRNA and protein (Fig 3B and C), as well as the

increase in resting ER calcium levels and histamine-induced ER calcium release (Fig 3D and F). DMF also ameliorates the increased cytosolic calcium levels in both resting and histamine-stimulated conditions (Fig 3E and G). Although these results confirm that Nrf2 controls GPx8 expression and ER calcium homeostasis, Nrf2 regulates many other genes. Hence, we sought to demonstrate the specific role of GPx8. To this end, GPx8 was overexpressed in HeLa cells (Fig 4A). As expected, GPx8 overexpression decreased ER calcium levels (Fig 4B) and histamine-induced ER calcium responses (Fig 4D). In Nrf2-silenced cells, GPx8 overexpression also lowered cytosolic

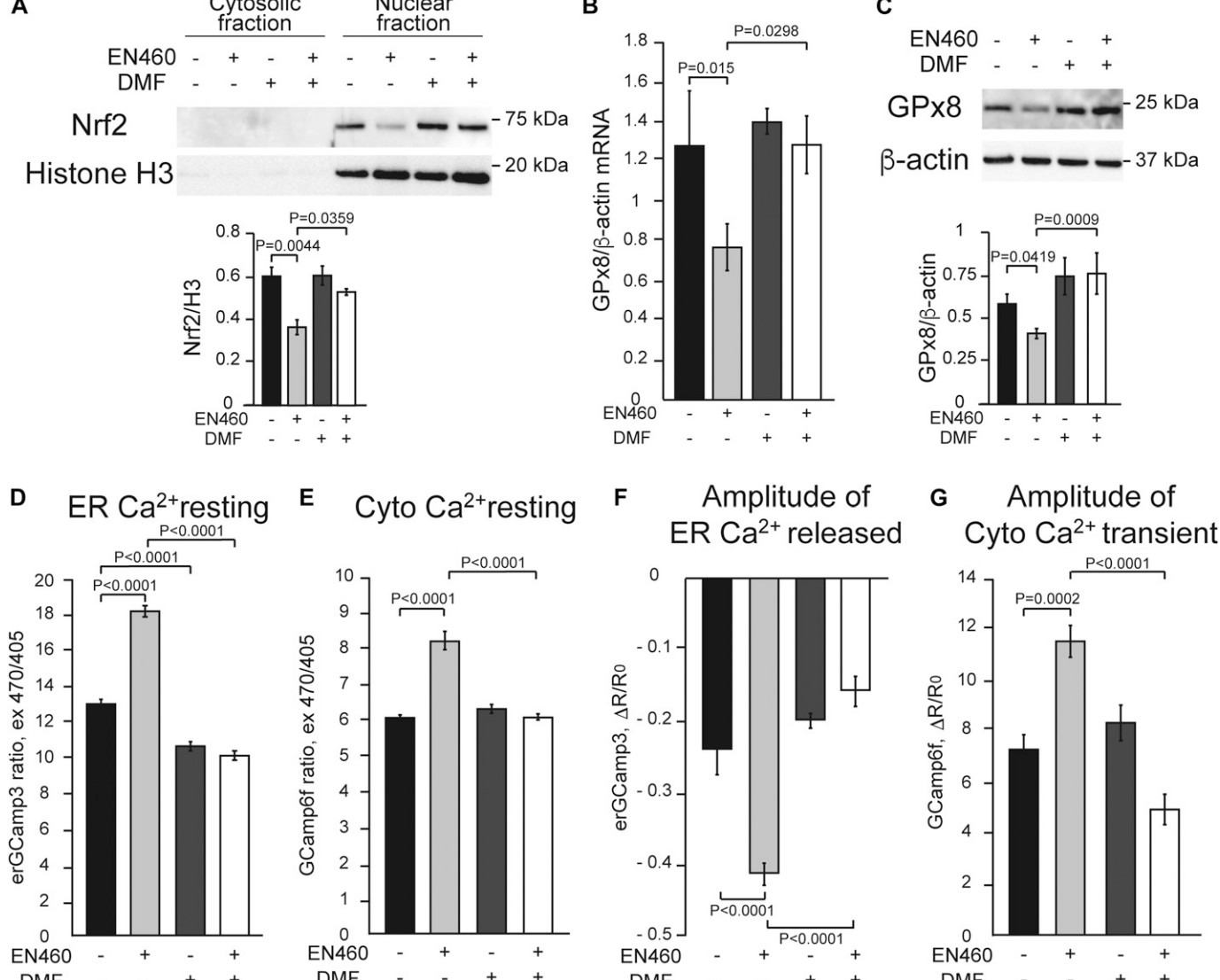

**Figure 3. Activation of Nrf2 with DMF prevents adverse effects of Ero1 inhibition on Nrf2 signaling and ER calcium dysregulation.**
HeLa cells were treated with 30 μM DMF (+) or vehicle (−) for 48 h together with 50 μM EN460 (+) or vehicle (−) for 90 min. All data are presented as mean ± SEM. **(A)** Representative Western blot of Nrf2 in the nuclear and cytosolic fractions and quantification of nuclear Nrf2 protein levels normalized by Histone H3. **(B)** GPx8 mRNA levels normalized by β-actin. **(C)** Representative Western blot and quantification of GPx8 protein, normalized by β-actin. For (A, B, C), n = 3. P values between indicated groups by paired one-way ANOVA test with Sidak's correction are shown. **(D)** Resting ER calcium content measured by erGCaMP3. n = 224, 240, 232, 242 cells from two sets of experiments. **(E)** Resting cytosolic calcium level measured by GCaMP6f. n = 200, 174, 181, 172 cells from two sets of experiments. **(F)** ER calcium released upon 100 μM histamine stimulation, as measured by the decline in erGCaMP3 fluorescence. n = 30, 32, 41, 33 cells from two sets of experiments. **(G)** Cytosolic calcium peak upon ER calcium release with 100 μM histamine, measured by GCaMP6f fluorescence flux. n = 30, 40, 30, 43 cells from two sets of experiments. For (D, E, F, G), P values between indicated groups assessed by Kruskal–Wallis test with Dunn's correction are shown.

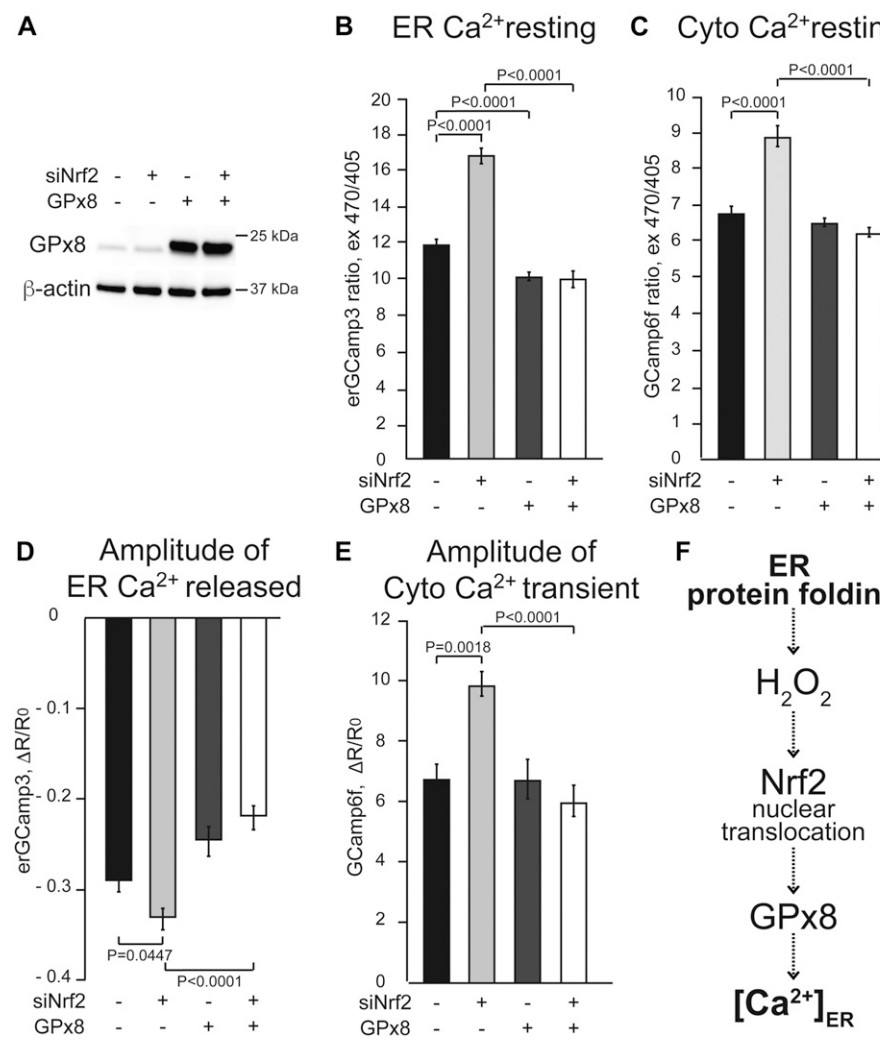

**Figure 4. Expression of GPx8 normalizes ER calcium dysregulation, downstream of Nrf2 signaling.**
HeLa cells were transfected with GPx8 (+) or pcDNA3 (−) for 48 h, and then with Nrf2 siRNA (+) or control siRNA (−) for 24 h. All data are presented as mean ± SEM. **(A)** Representative Western blot of GPx8 overexpression and $\beta$-actin as loading control. **(B)** Resting ER calcium measured by erGCaMP3. n = 317, 186, 253, 337 cells from three sets of experiments. P values between indicated groups by Kruskal–Wallis test with Dunn's correction are shown. **(C)** Resting cytosolic calcium levels measured with GCaMP6f. n = 325, 331, 505, 402 cells from three sets of experiments. P values between indicated groups by Kruskal–Wallis test with Dunn's correction are shown. **(D)** ER calcium released upon 100 $\mu M$ histamine stimulation, measured by drop in erGCaMP3 fluorescence. n = 36, 36, 52, 77 cells from three sets of experiments. P values between indicated groups by paired one-way ANOVA test with Sidak's correction are shown. **(E)** Cytosolic calcium transient peak induced with 100 $\mu M$ histamine, measured by GCaMP6f. n = 41, 49, 61, 73 cells from three sets of experiments. P values between indicated groups by Kruskal–Wallis test with Dunn's correction are shown. **(F)** Proposed model linking ER protein folding and ER calcium through $H_2O_2$ signaling, Nrf2 nuclear translocation, and GPx8 expression.

calcium at rest (Fig 4C) and after histamine stimulation (Fig 4E). These findings indicate that GPx8 is a main player in the regulation of ER calcium downstream of Nrf2. Collectively, these results reveal a signaling pathway linking ER oxidative protein folding with ROS-mediated Nrf2 regulation that controls ER calcium homeostasis and signaling through GPx8 (Fig 4F).

### Mutant SOD1-induced pluripotent stem cell (iPSC)–derived ALS astrocytes have reduced ER oxidative protein folding and $H_2O_2$ levels

Alterations in ER redox protein folding (Parakh et al, 2013) and calcium homeostasis (Tadic et al, 2014; Vardjan et al, 2017; Bano & Ankarcrona, 2018) have been associated with a number of pathologies, including neurodegenerative disorders. Therefore, we wanted to investigate the pathophysiological relevance of the oxidative protein folding-Nrf2-ER calcium axis. We chose, as a neurodegenerative disease model, human astrocytes harboring a genetic mutation causative of familial ALS for three reasons. First, astrocytes express higher levels of Nrf2 and ARE-responsive genes than neurons (Shih et al, 2003; Liddell, 2017). Second, the role of astrocytes in the

non-cell autonomous death of motor neurons in ALS has been firmly established in animal and cellular models of the disease (reviewed in Lee et al (2016)). Third, we had previously shown that intracellular calcium signaling is dysregulated in astrocytes from G93A SOD1 mutant mice, causing alterations of calcium-dependent secretory processes that impair motor neuron survival (Kawamata et al, 2014).

We used human iPSC-derived astrocytes (iPSAs) derived from a healthy control donor iPSC line and the isogenic mutant G93A SOD1 (G93A) line generated by targeted mutagenesis. By immunofluorescence, both control and G93A iPSA lines expressed the astrocyte markers, aldehyde dehydrogenase 1 family L1 (Aldh1L1), aquaporin 4 (Aqp4), and glial fibrillary acidic protein (GFAP) (Fig S2A). By Western blot, we found similar levels of ER calcium channel IP₃R2 and the metabotropic glutamate receptors mGluR 1/5 and mGluR 2/3, in mutant and in control iPSAs (Fig S2B and C). However, G93A cells showed increased GFAP expression, reduced glutamate transporters (Fig S2C), and accumulation of misfolded SOD1 (Fig S2D), suggesting that these cells recapitulate key features of mutant SOD1 astrocytes (Rothstein et al, 1995; Bristol & Rothstein, 1996; Schiffer et al, 1996; Hall et al, 1998).

We tested the hypothesis that mutant iPSAs have altered Ero1-dependent ER oxidative protein folding, by following the oxidation of intramolecular disulfide bonds in the J chain subunits of the polymeric immunoglobulins (JcM) in the ER. G93A iPSAs had a slower rate of JcM oxidation compared with control, as indicated by a delayed rate of decline in reduced JcM (Fig 5A). This result suggests inefficient folding capacity by Ero1 in mutant astrocytes. To assess whether impaired oxidative ER protein folding led to an unfolded protein response (UPR), we measured the levels of grp78/Bip and PDI, which are typically up-regulated during UPR. Surprisingly, in mutant iPSAs, these proteins were significantly down-regulated compared with controls (Fig S2E), suggesting that oxidative protein folding delay in mutant SOD1 iPSAs does not result in UPR. Furthermore, it suggests that PDI down-regulation could participate in impaired protein folding.

Because the cysteine residues of SOD1 participate in its folding through intramolecular disulfide bonds and are involved in misfolding and aggregation of mutant SOD1 (Sheng et al, 2012), we investigated the role of SOD1 redox state in ER protein folding impairment. To this end, we compared the effects of overexpressing G93A SOD1, G93A with all four cysteines mutated to serine (G93A 4cys mut), and wild-type (WT) SOD1, on JcM oxidative folding. We found that JcM folding was delayed by G93A, but not by G93A 4cys mut, relative to WT SOD1 (Fig 5B), indicating that G93A SOD1 redox state drives ER protein folding impairment.

SOD1 is an abundant cytosolic protein but can also localize to the lumen of organelles, including the ER (Urushitani et al, 2008). Therefore, to dissect the role of SOD1 localized in the ER in modulating ER protein folding, we expressed WT, G93A, and G93A 4cys mut selectively targeted to the ER lumen. Interestingly, neither ER-targeted G93A nor ER-targeted G93A 4cys SOD1 caused JcM oxidative folding impairment, relative to ER-targeted WT SOD1 (Fig 5C). Taken together, these results suggest that G93A SOD1 impairs ER oxidative protein folding from the cytosol and not from within the ER lumen, and that the redox state of mutant SOD1 cysteines plays a critical role in altering ER protein folding. We predicted that delayed oxidative folding in G93A iPSAs decreases the production of $H_2O_2$ in the ER. Therefore, we assessed $H_2O_2$ levels with HyPer targeted to the ER or cytosol and found that G93A iPSAs had lower basal $H_2O_2$ in both compartments compared with controls (Fig 5D and E). To determine the effect of G93A SOD1 on cytosolic $H_2O_2$ levels of neurons, we used primary cortical neurons from the G93A SOD1 expressing mice (Hall et al, 1998) and compared them with neurons from non-transgenic littermates. In contrast to G93A iPSAs, in mutant SOD1 mouse primary neurons, HyPer fluorescence revealed higher cytosolic $H_2O_2$ levels, as compared with non-transgenic control neurons (Fig 5F), suggesting that G93A SOD1 astrocytes and neurons differ in their redox status.

Next, we assessed the ER redox state by measuring the activity of peroxiredoxin 4 (Prdx4). Prdx4 is an ER-resident peroxidase that was shown to function as a molecular redox sensor of ER $H_2O_2$ and to negatively regulate the localization of transmembrane proteins, such as glycerophosphodiester phosphodiesterase 2 (GDE2) (Yan et al, 2015). Therefore, the localization of GDE2 can be used as a marker of ER redox state. We found that the proportion of cells in which exogenously expressed recombinant GDE2-FLAG was localized at the plasma membrane was higher in G93A iPSAs compared with

controls (Figs 5G and S2F). Taken together, these results suggest that in mutant iPSAs, Ero1-dependent oxidative protein folding is impaired and, as a result, less $H_2O_2$ is produced in the ER.

## The Nrf2 antioxidant response system is altered in G93A iPSAs

We assessed the effects of delayed ER oxidative protein folding and decreased $H_2O_2$ signaling on the Nrf2 antioxidant response in G93A iPSAs. Nuclear Nrf2 was decreased in these cells (Fig 5H), suggesting that Nrf2 signaling is repressed in mutant iPSAs. As a confirmation that decreased Nrf2 signaling in G93A iPSAs was due to insufficient $H_2O_2$, we found that when mutant iPSAs were treated with exogenous $H_2O_2$ (100 $\mu M$ for 30 min), GPx8 expression increased, indicating that Nrf2 signaling in these cells is repressed, but still functional (Fig S3A). To ensure that the decline in nuclear Nrf2 in mutant iPSAs was due to SOD1, we silenced SOD1 by ~70% (Fig S3B), which significantly increased nuclear translocation of Nrf2 (Fig S3C), suggesting that the alteration in Nrf2 nuclear translocation is dependent on the presence of SOD1, but is not the result of defective enzymatic activity of mutant SOD1. As expected, the downstream targets of Nrf2, GPx8 mRNA and protein levels (Fig 5I and J), NQO1, and HO-1 mRNA (Fig S3D and E), were decreased in G93A iPSAs compared with controls. In agreement with decreased GPx8, we found an increased rate of ER calcium uptake in mutant iPSAs (Fig 5K), consistent with higher SERCA activity.

## Intracellular calcium signaling is abnormal in G93A iPSAs

We then assessed baseline calcium levels and calcium dynamics in response to metabotropic stimuli in iPSAs. Consistent with increased SERCA activity, the ER of G93A iPSAs had elevated calcium content (Fig 6A) and enhanced ATP-induced ER calcium release, as shown by a steeper drop of erGCaMP3 fluorescence ratio (Fig 6B). Enhanced ATP-induced ER calcium release led to increased cytosolic calcium transients in G93A iPSAs (Fig 6C). In addition, G93A iPSAs had increased resting cytosolic calcium levels (Fig 6D), possibly because of enhanced calcium influx into the cells. To test this hypothesis, we measured store-operated calcium entry (SOCE) upon ER calcium depletion induced by the SERCA inhibitor thapsigargin (TG) and found that SOCE was increased in G93A iPSAs (Fig S4A). Last, to demonstrate that impaired ER protein folding is upstream of ER calcium dysregulation, we tested if the small molecule PDI mimetic dithiol, (+/−)-trans-1,2-bis(2-mercaptoacetamido)cyclohexane (BMC, 25 $\mu M$ for 18 h) could ameliorate the calcium alterations. BMC normalized ATP-induced ER calcium release in G93A iPSAs (Fig S4B), further supporting the mechanistic link between ER oxidative protein folding and calcium dysregulation.

Closely juxtaposed ER and mitochondria membranes form mitochondrial associated membranes, through which the two organelles exchange calcium ions. Therefore, to assess the effects of ER calcium alterations on mitochondrial calcium dynamics, we used GCaMP6f targeted to mitochondria. There was a small but significant decrease in resting mitochondrial calcium levels in mutant iPSAs compared with controls (Fig 6E). However, mitochondrial calcium uptake after ER calcium release induced by ATP was strongly increased in G93A iPSAs (Fig 6F). This evidence indicates that mutant iPSAs mitochondria take up more calcium as a result of increased ER calcium release.

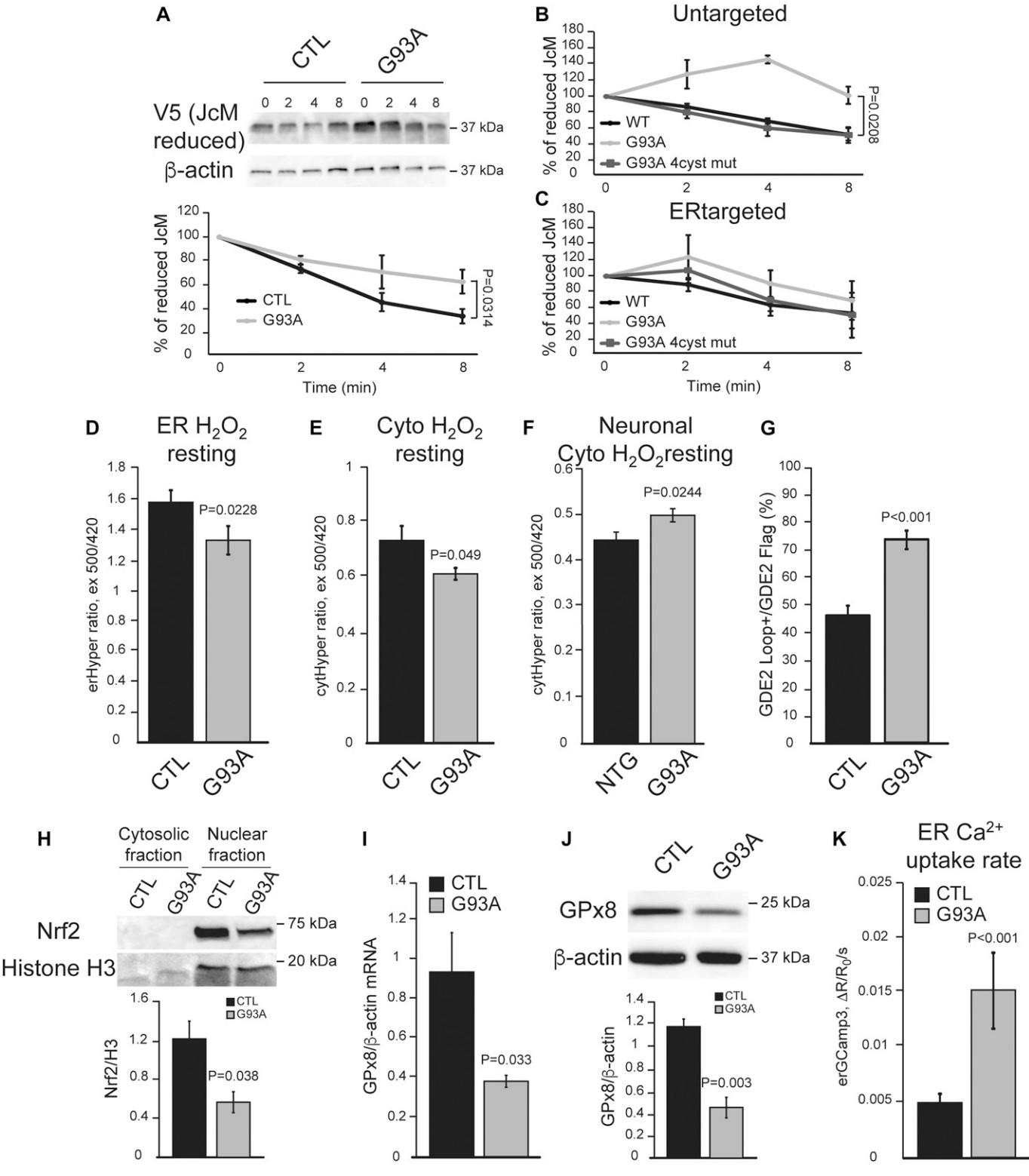

**Figure 5. ER oxidative protein folding is impaired and Nrf2 signaling diminished in G93A iPSAs.**
**(A)** Ero1-dependent protein folding assay after oxidation of the immunoglobulin protein JcM, tagged with V5. Top panel: representative Western blot of reduced JcM followed over 0–8 min of oxidation in control (CTL) and G93A iPSAs; bottom panel: quantification of reduced JcM levels, normalized by β-actin, and then to the fully reduced protein level at time 0. n = 3. P value by unpaired t test is indicated. **(B)** Quantification of reduced JcM levels in COS 7 cells expressing WT SOD1 (WT), G93A SOD1 (G93A), or G93A SOD1 with its four cysteine residues mutated to serine (G93A 4cyst mut). n = 3. P value by Mann–Whitney test is indicated. **(C)** Quantification of reduced JcM levels in COS 7 cells expressing ER targeted WT SOD1, G93A SOD1, or G93A 4cyst mut. n = 3. **(D)** Basal ER H2O2 content in CTL and G93A iPSAs measured by ER targeted

In primary cortical neurons from the G93A SOD1 expressing mice, resting ER calcium levels were unchanged relative to non-transgenic littermate neurons (Fig 6G), suggesting that G93A SOD1 astrocytes and neurons differ in their regulation of ER calcium homeostasis.

## Intracellular ATP levels and calcium-dependent vesicular secretion are increased in G93A iPSAs

Cytosolic calcium increase can up-regulate glycolysis (Schonekess et al, 1995), whereas calcium entry into mitochondria stimulates oxidative phosphorylation and ATP production through activation of calcium-dependent dehydrogenases (Griffiths & Rutter, 2009; Rizzuto et al, 2012). Because we found enhanced ER–mitochondria calcium transfer in mutant iPSAs, we sought to determine the effect of enhanced calcium signaling on bioenergetics. First, we used metabolic flux analysis to study mitochondrial respiration and glycolytic activities. We found that oxygen consumption rate (OCR) was increased in G93A iPSAs relative to control (Fig 6H and I). The extracellular acidification rate (ECAR), reflecting glycolytic production of lactate (Konrad et al, 2017), was not significantly increased in mutant iPSAs, as compared with controls (not shown). As a result, the OCR/ECAR ratio was higher in G93A iPSAs (Fig 6J), suggesting that these cells up-regulate oxidative metabolism. In agreement with increased oxidative phosphorylation, mitochondrial membrane potential was higher in mutant iPSAs than in controls (Fig 6K). On the other hand, the levels of respiratory chain subunits, the mitochondrial protein import translocator subunit Tom20, and the mitochondrial calcium uniporter showed no significant differences between G93A and control iPSAs (Fig S5A and B), suggesting that the altered bioenergetic profile in G93A iPSAs was not due to mitochondrial protein changes but to a calcium-dependent increase in mitochondrial respiration.

Next, we measured total intracellular ATP levels to assess the effect of increased calcium-dependent energy metabolism in G93A iPSAs. ATP at baseline was higher in mutant iPSAs than that in control (Fig 6L). Furthermore, G93A iPSAs had increased sensitivity to inhibition of glucose utilization because ATP content declined significantly more than in control cells when they were exposed to the hexokinase inhibitor 2-deoxyglucose (2DG, Fig 6L), indicating that the hypermetabolism in mutant iPSAs is driven by enhanced utilization of glucose. We also investigated ATP levels in subcellular compartments, using luciferase-based ATP reporters directed to the cytosol or to the ER. The sensitivity of these ATP reporters to variations in cellular ATP was confirmed by inhibition of glycolysis with 2DG (Fig S5C and D). In mutant iPSAs, we detected higher ATP-dependent luminescence, both in the cytosol and ER (Fig 6M and N). Because the ER does not have an intrinsic ATP generating machinery, this result reflects up-regulation of ATP import from the cytosol to the ER in G93A iPSAs.

It was shown that astrocytes from G93A SOD1 transgenic mice have up-regulated calcium-driven vesicular secretion, which negatively affects motor neuron viability (Kawamata et al, 2014). Therefore, we investigated if a similar phenotype is present in G93A iPSAs, measuring calcium-induced secretion of ATP into the medium by a luciferase-based assay. Upon induction of ER calcium release with TG, we detected enhanced secretion of ATP in G93A iPSA medium (Fig 6O). Collectively, these data suggest that mutant iPSAs have a higher calcium-induced bioenergetic output, which results in higher cellular ATP levels and enhanced calcium-dependent secretion.

## Nrf2 activation rescues GPx8 levels and ER calcium dysregulation in G93A iPSAs and prevents toxicity to motor neurons

Because we showed that Nrf2 activation by DMF corrects calcium dysregulation in HeLa cells exposed to Ero1 inhibition, we hypothesized that DMF could also rescue calcium dysregulation in G93A iPSAs. To this end, we treated iPSAs with DMF (30 µM for 48 h) and observed a strong increase in nuclear Nrf2 in G93A iPSAs (Fig 7A and B). Furthermore, DMF increased both mRNA and protein levels of GPx8 (Fig 7C and D). DMF treatment significantly decreased resting ER calcium content (Fig 7E), ATP-induced ER calcium release (Fig 7F), and of resting cytosolic calcium (Fig 7G), in G93A iPSAs. In agreement with the effects of DMF on intracellular calcium regulation, we also observed that DMF decreased calcium-dependent cell secretion from G93A iPSAs, but not control cells (Fig 7H).

To investigate the effects of Nrf2 activation on motor neuron viability, we exposed human iPSC-derived wild-type motor neurons to astrocyte conditioned medium (ACM). After challenging the motor neurons with two doses of ACM for 6 d, we evaluated motor neuron viability. Motor neurons in G93A iPSA ACM showed decreased viability compared with neurons in control ACM (Figs 8A, B, and G, and S6A and B). DMF ameliorated the toxicity of G93A iPSA ACM to motor neurons (Figs 8C and G, and S6C). Notably, DMF was protective only when added to astrocytes, and not when added to the ACM after it was harvested (Figs 8D and G, and S6D), indicating that DMF exerts its neuroprotective effect through astrocytes and not directly on motor neurons. Nrf2 silencing in G93A iPSAs further increased its toxicity to motor neurons (Figs 8E and G, and S6E), and this effect was ameliorated by GPx8 overexpression (Figs 8F and G, and S6F), further suggesting that DMF neuroprotection occurs through rescue of GPx8 expression in G93A iPSAs.

Together, these results indicate that in G93A iPSAs, the oxidative protein folding-Nrf2-ER calcium axis is impaired, leading to ER calcium dysregulation, increased cell secretion, and motor neuron toxicity. Activation of Nrf2 by DMF, in a ROS-independent manner, ameliorates the downstream ER calcium dysregulation, thereby decreasing calcium-dependent cell secretion and motor neuron toxicity (Fig 8H).

HyPer (erHyPer). n = 76, 57 cells from three sets of experiments. *P* value by Mann–Whitney test is indicated. **(E)** Resting cytosolic $H_2O_2$ levels in CTL and G93A iPSAs measured with HyPer (cytHyPer). n = 29, 62 cells from three sets of experiments. *P* value by Mann–Whitney test is indicated. **(F)** Resting cytosolic $H_2O_2$ levels in primary cortical neurons from non-transgenic (NTG) and G93A SOD1 (G93A) mice measured with HyPer. n = 52, 50 cells from two sets of experiments. *P* value by unpaired *t* test is indicated. **(G)** Quantification of the percentage of cells positive for surface translocated GDE2 (GDE2 Loop), over total GDE2 (GDE2 Flag). n = 54, 54 field acquired in two sets of experiments. *P* value by Mann–Whitney test is indicated. **(H)** Representative Western blot of Nrf2 in the nuclear and cytosolic fractions of CTL and G93A iPSAs. Quantification of nuclear Nrf2 normalized by H3 levels. n = 3. *P* value by unpaired *t* test is indicated. **(I)** GPx8 mRNA levels normalized by $\beta$-actin. n = 4. *P* value by unpaired *t* test is indicated. **(J)** Representative Western blot and quantification of GPx8 normalized by $\beta$-actin. n = 3. *P* value by unpaired *t* test is indicated. **(K)** ER calcium uptake rate based on SERCA activity in CTL and G93A iPSAs, measured by erGCaMP3. n = 33, 39 cells from five sets of experiments. *P* value by Mann–Whitney test is indicated.

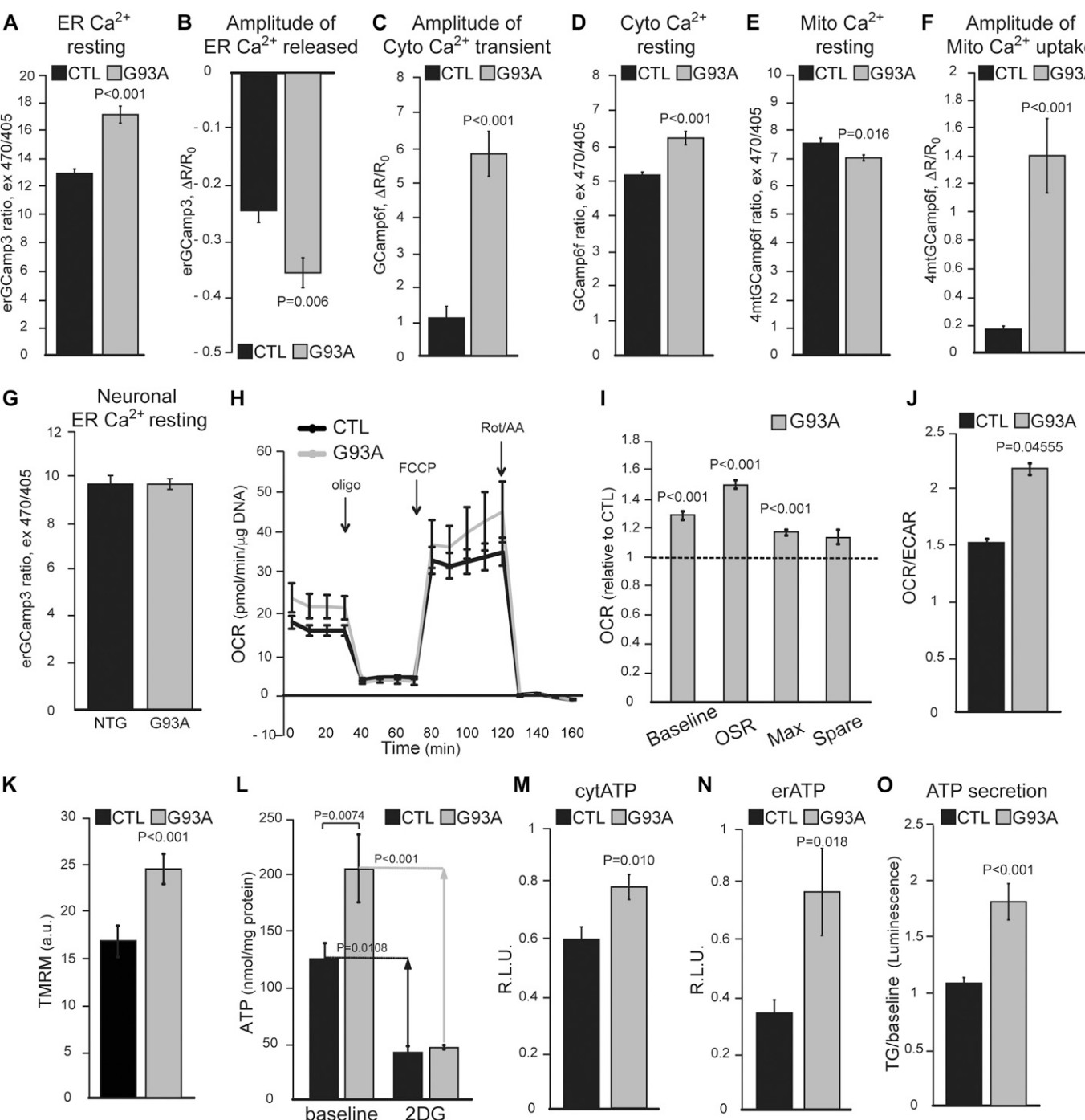

**Figure 6. ER calcium dynamics and energy metabolism are increased in G93A iPSAs.**
**(A)** Resting ER calcium levels in CTL and G93A iPSAs, measured with erGCaMP3. n = 207, 209 cells from two sets of experiments. **(B)** ER calcium released in CTL and G93A iPSAs upon ER calcium release stimulation with 100 $\mu$M ATP, measured as erGCaMP3 fluorescence drop. n = 24 cells for each genotype, from two sets of experiments. **(C)** Cytosolic calcium peak in CTL and G93A iPSAs upon 100 $\mu$M ATP stimulation, measured with GCaMP6f. n = 38, 37 cells from four sets of experiments. **(D)** Resting cytosolic calcium in CTL and G93A iPSAs measured with GCaMP6f. n = 220, 192 cells from two sets of experiments. **(E)** Resting mitochondrial calcium levels in CTL and G93A iPSAs, measured with GCamp6f targeted to mitochondria (4mtGCaMP6f). n = 195, 171 cells from three sets of experiments. **(F)** Mitochondrial calcium uptake in CTL and G93A iPSAs upon 100 $\mu$M ATP stimulation measured with 4mtGCaMP6f. n = 20, 20 cells in three sets of experiments. **(G)** Resting ER calcium levels in primary cortical neurons from non-transgenic (NTG) and G93A SOD1 (G93A) mice, measured with erGCaMP3. n = 76, 86 cells from two sets of experiments. In (A, B, C, D, E, F, G), P value determined by Mann–Whitney test is shown. **(H)** Representative curves of OCR in CTL and G93A iPSAs. Oligomycin (oligo), uncoupler FCCP (FCCP), rotenone and antimycin A (Rot/AA) were added at the indicated times. **(I)** Quantification of OCR metrics, normalized to CTL, including OCR baseline (Baseline), oligomycin sensitive rate (OSR), maximal respiration (Max), and spare respiratory capacity (Spare). **(J)** Average OCR/ECAR ratio determined from baseline OCR and ECAR values from each experiment. n = 15 wells from three individual experiments. P values by unpaired t test are indicated. **(K)** Mitochondrial membrane potential in CTL and G93A iPSAs assessed by the potentiometric dye TMRM. n = 60 cells from two individual experiments. P value by unpaired t test is indicated. **(L)** ATP content measured by a luciferase-based assay in CTL and G93A iPSAs, in glucose

# Discussion

In this study, we describe a novel signaling pathway that links two fundamental ER functions, oxidative protein folding and cellular calcium homeostasis. We show that these two processes are connected through $H_2O_2$ and Nrf2 signaling, which in turn regulate the expression of GPx8 to modulate ER calcium content. ER protein folding is highly sensitive to luminal calcium concentration, as many components of the protein folding apparatus are calcium-dependent, including the chaperones calnexin and calreticulin, PDI, and ERp57 (reviewed in Coe & Michalak (2009)). In light of our findings, we propose a homeostatic mechanism, which senses ER oxidative protein folding through ROS signaling and modulates ER calcium levels accordingly.

The involvement of GPx8 in regulating ER calcium homeostasis is supported by the observation that overexpression of GPx8 normalizes ER calcium, under Nrf2 silencing conditions. It was shown that GPx8 inhibition of SERCA activity requires both the ER transmembrane domain and the redox capability of GPx8 (Yoboue et al, 2017). As SERCA activity can be modulated by post-translational modification of cysteine residues, both on the intra- and extra-luminal portions of the protein (Li & Camacho, 2004; Thompson et al, 2014; Marino et al, 2015), it is likely that GPx8 modulates SERCA activity through redox modifications that remain to be identified. $H_2O_2$ produced by ER oxidative protein folding diffuses to the cytosol (Appenzeller-Herzog et al, 2016), where it provides signaling functions. However, when produced in excess, it can also cause oxidative damage to macromolecules. Thus, GPx8 could serve dual functions, controlling ER $H_2O_2$ output by regulating oxidative protein folding and $H_2O_2$ scavenging in the ER.

Here, we also show that impairment of the oxidative protein folding-Nrf2-ER calcium axis in G93A SOD1 astrocytes participates in toxicity to motor neurons. In the pathophysiological context of neurodegeneration, ROS are often thought to be detrimental, but in reality, the effects of ROS are likely to be context and cell-type specific, as their role in signaling in the CNS is well known (reviewed in Ren et al (2017)). Among CNS cell types, astrocytes have the highest Nrf2 expression (Shih et al, 2003; Bell et al, 2015) and play a major role in regulating the redox environment of neurons by supplying glutathione. Although the properties of ALS astrocytes that contribute to motor neuron toxicity are not yet fully understood, it was shown that increasing Nrf2 signaling ameliorates the phenotype of G93A SOD1 transgenic mice (Neymotin et al, 2011). It was also shown that Nrf2 was protective when expressed in astrocytes (Vargas et al, 2008), but not in neurons (Vargas et al, 2013), highlighting the specific benefit of astrocytic Nrf2-mediated neuroprotection.

Our findings suggest a mechanistic explanation of the neuroprotective effects of Nrf2 activation in G93A iPSAs. We propose that in these cells, the decrease in $H_2O_2$ is due to impaired ER oxidative protein folding. Although the precise contribution of ER-generated

$H_2O_2$ to the pool of cytosolic ROS in astrocytes remains to be further elucidated, we find that in HeLa cells, inhibition of Ero1 decreases cytosolic $H_2O_2$ and it is known that $H_2O_2$ leaks from the ER into the cytosol, spontaneously or with the help of GPx8 (Ramming et al, 2014) and aquaporins (Appenzeller-Herzog et al, 2016). In G93A iPSAs, decreased ER-derived $H_2O_2$ signaling blunts the expression of ARE enzymes, including GPx8. The involvement of redox signaling in altering ER peroxidases in mutant SOD1 iPSAs is further supported by the observation that exogenous $H_2O_2$ increase the expression of GPx8. Interestingly, cytosolic $H_2O_2$ and ER calcium in G93A SOD1 primary cortical neurons from transgenic mice did not reflect the changes observed in G93A iPSAs, suggesting that the pathway leading to altered ER calcium is not in common between G93A astrocytes and neurons. Nevertheless, as we show that HeLa cells exposed to Ero1 inhibitors can activate similar responses as iPSAs, one could extrapolate that the pathway is common to cell types which rely strongly on Nrf2 signaling, and that neurons are either incapable of putting a strong Nrf2-driven antioxidant response in place and accumulate ROS or that the ER protein folding-Nrf2-ER calcium axis is not as significant in neurons as it is in other cell types, especially astrocytes. In accord with this hypothesis, neurons express low levels of Nrf2 and antioxidant proteins (Shih et al, 2003).

What leads to ER protein folding impairment in mutant iPSAs? The finding that expression of mutant SOD1 lacking cysteines does not alter ER protein folding suggests that SOD1 cysteine residues are crucial in altering this process. Importantly, targeting mutant SOD1 directly to the ER lumen did not significantly alter ER protein folding, implying that mutant SOD1 localized outside of the ER is responsible for ER protein folding impairment. The precise molecular mechanism of ER protein folding delay in mutant iPSAs needs to be further investigated, but it may involve the impairment of the PDI-Ero1 system through post-translational modifications. For example, PDI inactivation by S-nitrosylation has been reported in cells expressing mutant SOD1, and ALS patients have increased PDI expression in the spinal cord, possibly as a compensatory effect (Walker et al, 2010; Jeon et al, 2014). In support of a role for PDI in causing impaired ER protein folding and calcium dysregulation, we show that the PDI-mimetic BMC improves ER calcium homeostasis in mutant iPSAs.

What are the consequences of altered ER calcium homeostasis in ALS astrocytes? The ER is the major calcium store in astrocytes, and alteration of ER calcium signaling can have severe consequences for cellular functions. ER calcium release induces secretion of small molecules, such as ATP. Notably, aberrant ATP secretion was associated with enhanced astrocyte toxicity to motor neurons (Kawamata et al, 2014). Secreted ATP can be metabolized by extracellular ectonucleotidases to adenosine, which has been shown to be toxic for motor neurons after binding to adenosine receptors (Ng et al, 2015). We also find that increased ER calcium signaling stimulates mitochondrial energy metabolism and ATP production, which could

medium (baseline) or 2-deoxyglucose (2DG) containing medium. Arrows indicate the difference between baseline and upon 2DG treatment. n = 6 wells from two individual experiments. *P* values between indicated groups by paired one-way ANOVA test with Sidak's correction are shown. **(M, N)** ATP content in the cytosol (M) and ER (N) in CTL and G93A iPSAs, measured with luciferase-based reporter plasmids. n = 8 wells from four sets of experiments. *P* value by unpaired *t* test is indicated. **(O)** Quantification of ATP secreted into the conditioned medium by CTL and G93A iPSAs, upon ER calcium release induction with 5 *μ*M thapsigargin (TG). n = 17 lysates from three sets of experiments. *P* value by Mann–Whitney test is indicated.

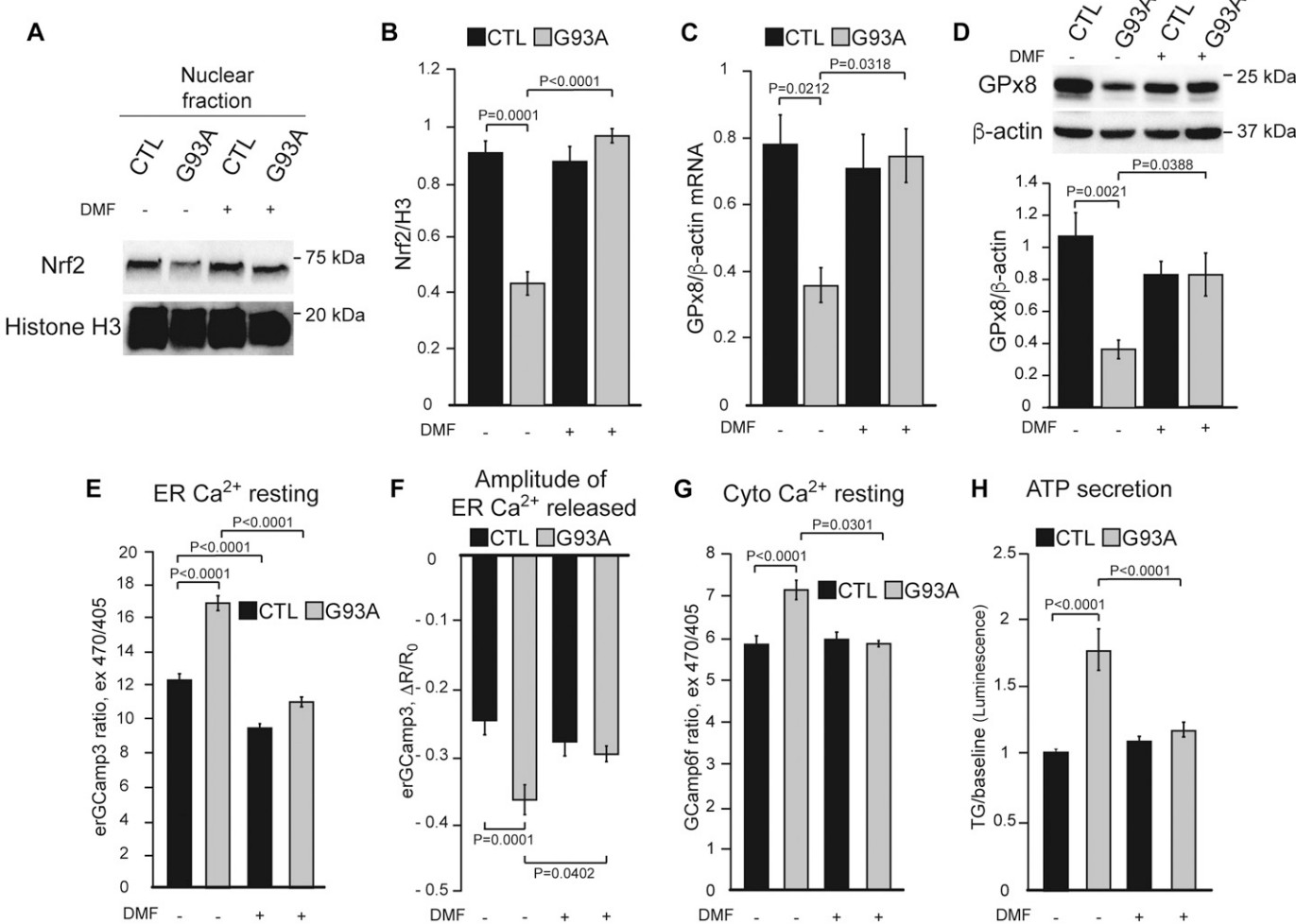

**Figure 7. Nrf2 activation by DMF restores ER calcium regulation in G93A iPSAs.**
CTL and G93A iPSAs were treated with 30 µM DMF (+) or vehicle (−) for 48 h. **(A)** Representative Western blot of nuclear Nrf2 from CTL and G93A iPSAs. **(B)** Quantification of nuclear Nrf2 normalized by H3. **(C)** GPx8 mRNA levels normalized by β-actin. **(D)** Representative Western blot and quantification of GPx8 levels, normalized by β-actin. In (B, C, D) n = 3. P values between indicated groups by paired one-way ANOVA test with Sidak's correction are shown. **(E)** Resting ER calcium levels in CTL and G93A iPSAs measured with erGCaMP3. n = 156, 187, 185, 163 cells from two sets of experiments. P values between indicated groups by Kruskal–Wallis test with Dunn's correction are shown. **(F)** ER calcium released with 100 µM ATP in CTL and G93A iPSAs, measured by erGCaMP3 fluorescence drop. n = 24, 24, 24, 26 cells from two sets of experiments. P values between indicated groups by paired one-way ANOVA test with Sidak's correction are shown. **(G)** Resting cytosolic calcium levels in CTL and G93A iPSAs, measured with GCaMP6f. n = 148, 174, 165, 174 cells from two sets of experiments. P values between indicated groups by Kruskal–Wallis test with Dunn's correction are shown. **(H)** ATP secretion into the conditioned medium by CTL and G93A iPSAs upon ER calcium release induced by 5 µM TG. n = 16 lysates from three sets of experiments. P values between indicated groups by paired one-way ANOVA test with Sidak's correction are shown.

increase the pool of ATP available for secretion. Therefore, modulation of ATP production by calcium could have both adaptive (i.e., meeting energy demands) and maladaptive (i.e., toxic secretory) effects.

In conclusion, this study reveals a novel mechanism of interplay between ER protein folding, ROS-Nrf2 signaling, and ER calcium homeostasis. This hormetic signaling maintains the balance between oxidative protein folding and ER calcium. We propose that alterations of this axis participate in motor neuron toxicity of G93A SOD1 astrocytes. Whether these interconnected pathways can be targeted for therapy in human ALS remains to be determined, but some encouraging evidence exists in animal models of the disease (reviewed in Liddell (2017)). Here, we show that treatment of G93A iPSAs with DMF, a drug that is currently in use for patients affected

by multiple sclerosis (Linker & Gold, 2013), ameliorates motor neuron death in a co-culture system. This evidence provides a mechanistic understanding of SOD1 ALS and the potential for more targeted therapeutic approaches.

# Materials and Methods

### Reagents

Unless otherwise noted, all chemical reagents were purchased from Sigma-Aldrich and all tissue culture reagents were purchased from Thermo Fisher Scientific.

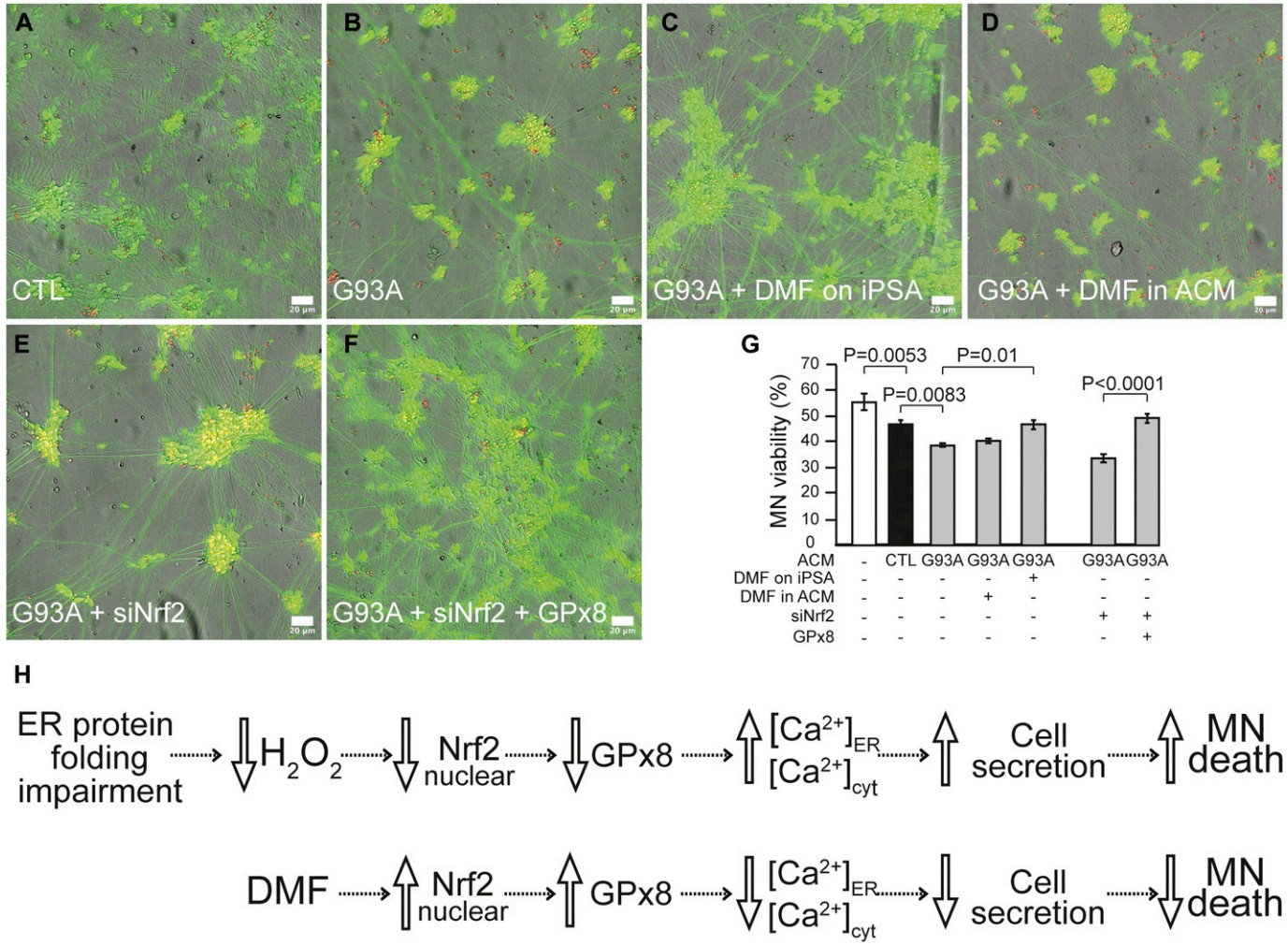

**Figure 8. Nrf2 activation restores GPx8 levels in G93A iPSAs and prevents motor neuron toxicity.**
Motor neurons cultured for 6 d in ACM were loaded with calcein (green) and PI (red) for 10 min and imaged. **(A, B, C, D, E, F)** Representative images of merged channel acquisitions (red, green, and bright field) of motor neurons cultured in ACM from the following conditions: untreated CTL (A), untreated G93A (B), DMF treated G93A (C), DMF added to G93A ACM (D), Nrf2 silenced G93A (E), and GPx8 overexpressed/Nrf2 silenced G93A (F). For single channel representative images, see Fig S6. The scale bars represent 20 $\mu$m. **(G)** Quantification of motor neuron viability, expressed as the percentage of live cells over the total number of motor neurons cultured in the indicated conditions. n = 9 wells for each condition from three sets of experiments. P values between indicated groups were determined by paired one-way ANOVA test with Sidak's correction. **(H)** Proposed scheme of the ER protein folding-Nrf2-ER calcium axis impairment in G93A SOD1 astrocytes, its involvement in motor neuron toxicity, and the protective effects of DMF.

## Cell culture

HeLa and COS 7 cells were grown in DMEM (high glucose, pyruvate) supplemented with 10% FBS and penicillin–streptomycin. The cells were transfected with a standard calcium phosphate procedure.

iPSAs were purchased from Cellular Dynamics International (CDI). The control astrocytes are differentiated from iPS cells derived from a healthy control donor (iCell astrocytes). The G93A SOD1 line is from the CDI MyCell Disease product (01434.751) and it is isogenic to the control line. iPSAs were maintained in DMEM (high glucose, pyruvate) supplemented with 10% FBS (Hyclone/GE Heatlhcare Bio-Sciences), N2 supplement, GlutaMAX, and penicillin–streptomycin and plated on Matrigel-coated (Corning) plates. The cells were split with TrypLE reagent to plate for experiments, and

assays were performed 5–7 d upon plating. iPSAs were used within DIV 17–35.

Wild-type iPSC-derived motor neurons (iCell motor neurons, CDI) were thawed and cultured according to the manufacturer's protocol. Motor neurons were plated on poly-D-lysine/Matrigel–coated plates, in a medium composed of iCell Neural base medium, iCell Neural supplement, and iCell Nervous System supplement (CDI).

Mouse primary cortical neurons from P0-2 newborn C57Bl/6J mice were isolated using a stereomicroscope and digested with trypsin at 37°C. After two digestion steps, the cells were counted and seeded on poly-L-lysine–coated glass coverslips. Neurons were cultured in MEM supplemented with 10% horse serum, N2 supplement, B27 supplement, sodium pyruvate, biotin, glucose, L-glutamine, and penicillin–streptomycin for five DIV before transfection.

## Cell treatments, transfections, and gene silencing

HeLa cells were treated with 50 $\mu$M Ero1 Inhibitor II-EN460 (Millipore-Merck), prepared in DMSO for 90 min. HeLa cells and iPSAs were treated with 30 $\mu$M DMF (dissolved in DMSO) for 48 h. 25 $\mu$M BMC (dissolved in methanol; Cayman Chemical) was applied to iPSAs overnight for 18 h. iPSAs were treated with 100 $\mu$M hydrogen peroxide ($H_2O_2$) for 30 min before the assay.

iPSAs were transfected with FugeneHD (Promega) 48 h before performing assays, following the manufacturer's protocol, using a 1:3 DNA ($\mu$g) to FugeneHD ($\mu$l) ratio. Neurons were transfected with Lipofectamine 2000, according to the manufacturer's instruction and experiments were performed 24 h later.

For gene silencing, we used endoribonuclease-prepared siRNA pools (MISSION esiRNA; Sigma-Aldrich), composed of a heterogeneous mixture of siRNAs all targeting the same mRNA sequence. To silence human Nrf2 and SOD1, we used #EHU093471 and #EHU050511, respectively, and #EHURLUC, targeting Renilla Luciferase, was used for the negative control. The cells were transfected with MISSION siRNA transfection reagent, according to the manufacturer's protocol, for 24 h (Nrf2) or 48 h (SOD1) before the assays.

## Plasmids

GPx8 ORF in pcDNA3.1$^+$/(K)DYK (oHu31975) was purchased from GenScript. ER-targeted SOD1 plasmids were created by adding restriction enzymes SalI and NotI to the cytosolic wild-type, G93A, and G93A C6S/C57S/C111S/C146S SOD1 ORF (Kawamata & Manfredi, 2008) by standard PCR, and cloning into the pCMV/myc/ER vector (Thermo Fisher Scientific). ER targeting of SOD1 was tested by immunocytochemistry with anti-myc antibody and positive co-localization with the ER resident calreticulin (data not shown). For ATP assessment in the ER, a luciferase ATP reporter (Syn-ATP [Rangaraju et al, 2014], #51819; Addgene), targeted to the ER lumen (erATP), was created by the addition of a calreticulin targeting motif at the N terminus and a KDEL retention signal at the C terminus of luciferase by PCR. The PCR product was cloned into the pTARGET vector (Promega) following the manufacturer's protocol. All plasmids were confirmed by sequencing.

## Live cell calcium imaging

Basal cytosolic, mitochondrial, and ER calcium levels were determined by the ratiometric fluorescence measurements using GCaMP6f (plasmid #40755; Addgene), GCaMP6f targeted to mitochondria (4mtGCaMP6f [Patron et al, 2019]), or GCaMP3 targeted to the ER (erGCaMP3, plasmid #63885; Addgene), respectively. After 48 h of transfection, the cells were mounted in a custom made, open-bath imaging chamber and maintained in Krebs-Ringer modified buffer (KRB, in mM: 135 NaCl, 5 KCl, 1 MgSO$_4$, 1 MgCl$_2$, 0.4 KH$_2$PO$_4$, 20 Hepes, and 1 CaCl$_2$, pH = 7.4). Imaging was performed on an Olympus IX83 inverted fluorescence microscope equipped with a 40× objective lens (N/A 0.90). The cells were alternately excited at 470 and 405 nm and fluorescence was collected through a 516-nm band-pass filter. Exposure time was set to 400 ms (for cytosolic and mitochondrial measurements) or 600 ms (for ER measurements) at 470 nm and to 200 ms at 405 nm. Each field was acquired for 10 s (1 frame/s).

For the measurements of ER calcium release, cytosolic calcium flux, and mitochondrial calcium uptake, the cells were transfected with the same calcium probes and imaging was performed as described above. The cells were stimulated with 100 $\mu$M histamine (HeLa) or 100 $\mu$M ATP (iPSA) by perfusion to release ER calcium through IP$_3$ receptors, and fluorescence change was monitored for 5 min. The experiments were terminated by lysing the cells with 100 $\mu$M digitonin in a hypotonic solution containing 10 mM calcium.

To measure SERCA activity, cells transfected with erGCaMP3 were treated with 1 $\mu$M ionomycin (calcium ionophore) for 5 min, perfused with 100 $\mu$M EGTA for 1 min, followed by KRB containing 1 mM CaCl$_2$ for 5 min to induce ER calcium uptake. When fluorescence reached a plateau, the cells were perfused with 1 $\mu$M thapsigargin (TG) for 3 min to monitor the passive leak from the ER. The experiments were terminated by lysing the cells with 100 $\mu$M digitonin in a hypotonic solution with 10 mM calcium. The ER calcium uptake rate was determined by the first derivative of the first 30 s of calcium influx.

Cytosolic calcium transient assessment by Fluo-4 was performed as described previously (Kawamata et al, 2014). Time-lapse images were acquired every 2 s on the Leica TSC SP5 confocal microscope with a water immersion 20× objective lens (N/A 0.70), with temperature control settings for stage and perfusion. ER calcium was released with 20 $\mu$M ATP in calcium-free buffer for the BMC experiments, and fluorescence change monitored over 6 min. To measure SOCE, the cells were perfused with 1 $\mu$M TG in calcium-free buffer for 6 min, followed by 2 mM calcium-containing buffer for 9 min. SOCE was determined by the difference between the baseline upon ER calcium depletion with TG and the cytosolic peak induced by external calcium.

## H$_2$O$_2$ measurement with HyPer

Cells were transfected with HyPer (cytHyPer, #42131; Addgene) or ER-targeted HyPer (erHyPer [Enyedi et al, 2010]) to assess redox states in the cytosol and ER lumen, respectively. After 48 h of transfection, the cells were mounted in an open-bath custom-made imaging chamber and maintained in KRB. Imaging was performed on the Olympus IX83 inverted fluorescence microscope on the 40× objective lens (N/A 0.90). The cells were excited with 500 and 420 nm, and fluorescence was collected through a 516-nm band-pass filter. Fluorescence was recorded every 10 s at baseline for 2 min, and then 10 mM DTT was added to fully reduce the probe.

## Image analysis

Fluorescence analysis was performed with the Fiji distribution of ImageJ (Schindelin et al, 2012). Background fluorescence was corrected by subtracting mean pixel values of a cell-free region of interest from every frame.

## RNA extraction, reverse transcription, and quantitative real-time PCR

For expression analysis of GPx8, HO-1, and NQO1, total RNA was extracted from iPSAs using the SV Total RNA Isolation Kit (Promega),

following the manufacturer instructions. RNA was quantified with an Eppendorf Bio Plus photometer. Using an equal amount of RNA from each sample, cDNA was generated with ImProm-II Reverse Transcription System (Promega) and analyzed by real-time qPCR using SYBR green chemistry (Thermo Fisher Scientific). Real-time PCR standard curves were constructed with serial dilutions of cDNA from analyzed samples using at least five dilution points. The efficiency of all primer sets was between 95 and 105%. $\beta$-actin was used as an internal control for cDNA quantification and normalization of the amplified products. Primer sequences for NQO1 and HO-1 were described previously (Makabe et al, 2010; Li et al, 2014). The GPx8 primers were designed and analyzed with Primer3. Primer sequences were as follows:

hs-GPx8-fw: TGCAGCTTACCCGCTAAAAT
hs-GPx8-rv: ATGACTTCAATGGGCTCCTC
hs-$\beta$Actin-fw: ATAGCACAGCCTGGATAGCAACGTAC
hs-$\beta$Actin-rv: CACCTTCTACAATGAGCTGCGTGTG.

## GDE2 immunocytochemistry

iPSAs were transfected with mGDE2-FLAG (Yan et al, 2015). 48 h post transfection, the cells were incubated with chlorpromazine in Hank's balanced saline solution for 30 min at 4°C, followed by deglycosylation with PNGase F at 37°C for 1 h. After the cells were washed with PBS, they were incubated with an anti-GDE2 loop antibody (1:200) at 37°C for 1 h. Cy2-conjugated antirabbit secondary antibody (1:500) was applied at room temperature for 45 min. The cells were then fixed with 4% PFA for 10 min, followed by permeabilization, blocking, and staining with anti-Flag antibody (1:200; Sigma-Aldrich) and Cy3 conjugated antimouse secondary antibody (1:500). Upon imaging on the Leica TCS SP5 confocal microscope, the number of transfected cells with positive staining by the GDE2 loop antibody on the surface, as well as Flag antibody staining on the surface and internally in the cytosol was counted. Data are presented as percentage of cells GDE2 Loop+/GDE2 Flag+ per field.

## Cellular fractionation

Nuclear and cytoplasmic extraction was performed with the NE-PER Nuclear and Cytoplasmic Extraction kit, following the manufacturer's instructions. Briefly, the cells were harvested and lysed in CER I and CER II buffers and centrifuged at 16,000$g$ for 5 min to obtain the cytosolic fraction. The insoluble pellet fraction was lysed in NER buffer and centrifuged at 16,000$g$ for 10 min to obtain the nuclear fraction. Protein concentration was determined by standard BCA protein assays. Nuclear and cytosolic protein contents were analyzed by Western blot.

## Western blots

Cells were lysed in RIPA buffer containing 25 mM Tris, pH 7.4, 125 mM NaCl, 1 mM EGTA, 1% Triton X-100, 0.5% sodium deoxycholate, 0.1% sodium dodecyl sulfate, phosphatase inhibitor, and protease inhibitor cocktail (Roche Diagnostics) for 40 min on ice and centrifuged at 16,000$g$ for 20 min at 4°C. Protein concentration was determined by standard BCA protein assays.

Western blots were performed on the Bio-Rad Western flow system. 30 $\mu$g of proteins (heated at 95°C for 10 min) were loaded on AnykDa precast gels (Bio-Rad). Proteins were transferred onto nitrocellulose membranes, blocked in 5% milk/TBS-T, immunoblotted with appropriate primary and horseradish peroxidase-conjugated secondary antibodies (1:5,000; Jackson ImmunoResearch), and visualized with chemiluminescent substrates on the Bio-Rad Chemidoc Touch. The membranes were stripped using BlotFresh Western Blot Stripping reagent (SignaGen Laboratories). All primary antibodies were used at 1:1,000. EAAT1, OXPHOS cocktail (Abcam), grp78/Bip (BD Transduction Lab), MCU, $\beta$-actin (Sigma-Aldrich), V5-tag (Thermo Fisher Scientific), Tom20, Nrf2 (SCBT), Histone H3 (Cell Signaling Technology), IP$_3$R2 (Chemicon International) mGluR 1/5 (clone number N75/33 Antibody Registry ID P31424; UC Davis/NIH NeuroMab facility), mGluR 2/3 (PhosphoSolutions), GFAP (Dako/Agilent), SOD1, PDI (Enzo Life Sciences), GPx8 (AdipoGen), EAAT2 (gift of Dr. D Trotti, Thomas Jefferson University), and SOD1 C4F6 (MEDIMABS). For further details, see the antibody list in Table S1.

## Immunocytochemistry

All steps except for primary antibody incubation were performed at room temperature. The cells were fixed with 4% PFA for 10 min. Upon three washes in PBS, cells requiring permeabilization were treated with 0.3% Triton X-100 (in PBS) for 10 min with gentle agitation. Cells that were not permeabilized were incubated with block solution upon fixation and washes. The cells were blocked with 1% BSA and 10% NGS (in PBS) for 1 h and incubated in appropriate primary antibodies in blocking solution overnight at 4°C. Upon three PBS washes, the cells were incubated in fluorophore-conjugated secondary antibodies (1:500; Jackson ImmunoResearch) for 1 h at room temperature. The cells were washed three times in PBS and mounted on slides for imaging with Fluoromount-G (Southern Biotech). Single-plane images were acquired on a Leica TCS SP5 confocal microscope equipped with a 63× objective (N/A 1.40). Primary antibodies used were Aldh1L1 (1:100, clone number N103-31, Antibody Registry ID P28037; UC Davis/NIH NeuroMab facility), Aqp4 (1:100; SCBT) and GFAP (1:500; Dako/Agilent).

## JcM oxidation assay

Cells were transfected with JcM-V5 plasmid (Enyedi et al, 2010). 48 h post transfection, the cells were treated with 10 mM DTT in DMEM for 20 min at 37°C to reduce disulfide bonds. The cells were washed in DMEM and incubated at 37°C between 0 and 8 min for disulfide reoxidation. Oxidation was stopped at appropriate time intervals by washing with ice-cold PBS containing 10 mM N-ethylmaleimide (NEM). The cells were harvested and lysed in buffer containing 1% Triton X-100, 50 mM Tris, pH 7.4, 150 mM NaCl, 10 mM NEM, and protease inhibitor cocktail for 30 min on ice. Cell lysates were cleared by centrifugation at 16,000$g$ for 20 min at 4°C. Proteins were suspended in loading dye without reducing agents, and a nonreducing SDS–PAGE was performed. Proteins were transferred onto nitrocellulose membranes and probed for V5-tag antibody, as well as $\beta$-actin following standard immunoblotting procedures.

## Mitochondrial stress test on seahorse XF96 flux analyzer

Mitochondrial respiration and ECARs were analyzed on the Seahorse XF96 flux analyzer (Agilent). The cells were plated on the XF 96-well culture plates (Agilent) 5 d before the assay. On the day of the assay, growth medium was replaced with 200 $\mu$l of XF Assay Medium (Agilent) supplemented with 5 mM glucose, 1 mM pyruvate, and 4 mM glutamine prewarmed at 37°C, without $CO_2$ for 1 h. Mitochondrial stress test assays to record OCRs and ECARs were performed with appropriate inhibitors, including 1 $\mu$M oligomycin, 2 $\mu$M FCCP, and 0.5 $\mu$M antimycin/0.5 $\mu$M rotenone, as described previously (Konrad et al, 2017). OCR and ECAR values were normalized by cellular DNA content. Based on OCR measurements, we assessed the following parameters: baseline respiration by intact cells in medium containing glucose, glutamine, and pyruvate, oligomycin-sensitive respiration (OSR), representing the ATP generating respiration, FCCP-induced, maximal respiration (Max) for uncoupled oxygen consumption, and spare respiratory capacity (Spare) determined by the difference between maximal and baseline respiration. Experiments were performed on a minimum of five technical replicate wells per assay.

## Mitochondrial membrane potential measurements

Tetramethylrhodamine methyl ester (TMRM; Thermo Fisher Scientific) was used to assess mitochondrial membrane potential. Cells were loaded with 10 nM TMRM for 30 min, and imaged live in Hank's buffered saline solution containing TMRM. Fields of cells were imaged at 543/590 nm ex/em on the Leica TCS SP5 confocal microscope with the 63× objective (N/A 1.4). The cells were imaged again upon addition of 1 $\mu$M FCCP, to dissipate membrane potential and obtain non-membrane potential dependent fluorescence for background subtraction. Average fluorescence intensity per cell was measured by Metamorph (Molecular Devices).

## Intracellular ATP content measurements

Intracellular ATP content in iPSAs was measured by a luciferase/luciferin–based assay, as described previously (Konrad et al, 2017). The cells were plated in replicates of nines on 96-well tissue culture plates 5 d before the assay. On the day of the assay, triplicate wells were incubated with either DMEM containing 5 mM D-glucose, 4 mM glutamine, and 1 mM pyruvate (baseline) or DMEM with 5 mM 2-deoxyglucose (2DG) instead of glucose, for 90 min. After incubation, the cells were washed and lysed in 2.5% (wt/vol) trichloroacetic acid on ice, and luminescence emitted by addition of luciferase/luciferin was measured on the SpectraMax plate reader (Molecular Devices). ATP content was normalized to cellular protein content.

## Subcellular ATP content assessment

The luciferase ATP reporters (cytATP [Gajewski et al, 2003] or erATP) were transfected into cells. The reporter response to ATP synthesis blockage through glycolysis was tested by treating the transfected cells with 5.5 mM 2DG instead of D-glucose in the culture medium for 1 h at 37°C. Luciferase activity was recorded on the SpectraMax plate reader (Molecular Devices) upon addition of luciferin. The

cells were permeabilized with 100 $\mu$M digitonin, in the presence of excess ATP, luciferin, and $MgCl_2$ to obtain maximal luciferase activity. Luciferase activities in samples were normalized to the maximal activities.

## Calcium-dependent cell secretion assay

ATP released from iPSAs was measured using the ENLITEN ATP reagent (Promega), as described previously (Kawamata et al, 2014). The cells were plated on 24-well plates, washed twice in DMEM, and incubated in DMEM for 30 min (Experiment Fig 6O) and for 15 min (Experiment Fig 7H) at 37°C, 5% $CO_2$. Secreted ATP in the conditioned medium was measured before and after the addition of 1 $\mu$M TG (5 min, for Fig 6O and 4 min for Fig 7H). Equal volumes of conditioned medium and luciferin/luciferase reagents were mixed and ATP content was immediately measured on the OPTOCOMP1 luminometer (MGM Instruments).

## Motor neuron survival in ACM

1.5 × 10$^4$ iPSAs/well were plated on 24-well plates in the iPSA medium overnight, and then switched to motor neuron medium for 48 h. The cells were then cultured in fresh motor neuron medium (CDI) containing 30 $\mu$M DMF (or vehicle) for 48 h. For ACM with GPx8 expression, iPSAs were transfected with GPx8 (or pcDNA3) for 48 h and with siRNA for Nrf2 during the last 24 h. ACM was collected and centrifuged at 1,200$g$ for 5 min to remove cell debris. Freshly harvested ACM was used on day 2 of motor neuron culture, and the rest was frozen for use on day 5. For the condition to test if DMF directly affects motor neuron viability, we added 30 $\mu$M DMF to G93A ACM, after media collection (G93A+DMF in ACM) before addition to motor neuron culture.

iCell motor neurons were plated on 96-well plates in motor neuron medium with 3.8 × 10$^4$ cells/well. 2 d post-plating, 75% of medium was replaced with fresh motor neuron medium containing 25% ACM. At day 5, 75% of medium was replaced for the second time, with fresh motor neuron medium containing 25% ACM. At day 8, motor neurons were incubated with calcein AM, propidium iodide (PI), and Hoechst diluted 1:1,000 in culture medium for 10 min at room temperature. FITC, TRITC, DAPI, and phase-contrast images of motor neurons were acquired with the ImageXpress Pico Automated Imaging System equipped with a 20× objective (NA 0.40), in a 5% $CO_2$ and 37°C controlled environment (Molecular Devices). Images were processed using the Fiji distribution of ImageJ (Schindelin et al, 2012). Background was subtracted using the rolling ball algorithm from each channel separately. Nuclei were identified by thresholding the Hoechst channel and filtering objects for appropriate size and circularity. For each nuclear object identified, average pixel intensities of calcein and PI were measured. Data processing was done in python. Nuclear objects with average pixel intensities below manually selected thresholds (100 and 200 intensity value, for PI and calcein, respectively), presumably corresponding to false nuclei identification, were excluded from the analysis (4.55% of the objects excluded), which correspond to the missing data point within the origin of the ordinate and abscissa axis (i.e., white rectangle in Fig S6G). Data are expressed as percentage of

calcein-loaded live cells on the total number of cells (calcein and PI positive cells).

## Statistical analyses

All data are presented as mean ± SEM. Normality of the data was investigated by Shapiro–Wilk test and statistical significance was calculated accordingly using GraphPad. Statistical analysis information is indicated in the figure legends and *P* values in the figures.

# Supplementary Information

# Acknowledgements

We are grateful to Dr. Shanthini Sockanathan (Johns Hopkins University, Baltimore, MD) for providing the GDE2 plasmid and GDE2 loop antibody used in this study. We also thank Dr. Miklos Geiszt (Semmelweis University, Budapest, Hungary) for the ER-targeted HyPer and JcM constructs. This work was supported by grants from the NIH/NINDS R21NS104520 (to H Kawamata and G Manfredi) and R01NS062055 (to G Manfredi).

## Author Contributions

V Granatiero: conceptualization, data curation, formal analysis, investigation, methodology, and writing—original draft, review, and editing.
C Konrad: data curation, formal analysis, investigation, and methodology.
K Bredvik: data curation, formal analysis, and methodology.
G Manfredi: conceptualization, resources, formal analysis, supervision, funding acquisition, investigation, project administration, and writing—original draft, review, and editing.
H Kawamata: conceptualization, data curation, formal analysis, supervision, funding acquisition, investigation, methodology, project administration, and writing—original draft, review, and editing.

## Conflict of Interest Statement

The authors declare that they have no conflict of interest.

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
