## [Reviewer comments · Life Science Alliance]

Nrf2 signaling links ER oxidative protein folding and calcium homeostasis in health and disease

Veronica Granatiero, Csaba Konrad, Kirsten Bredvik, Giovanni Manfredi and Hibiki Kawamata

DOI: 10.26508/lsa.201900563

Corresponding author(s): Prof. Hibiki Kawamata (Weill Cornell Medicine)

Review timeline:

Submission Date:	2019-09-24
Editorial Decision:	2019-09-25
Revision Received:	2019-10-16
Editorial Decision:	2019-10-18
Revision Received:	2019-10-20
Accepted:	2019-10-21

Scientific Editor: Andrea Leibfried

Transaction Report:

No Peer Review Process File is available with this article, as the authors have chosen not to make the review process public in this case.

1st Editorial Decision

25 September 2019

September 25, 2019

Re: Life Science Alliance manuscript #LSA-2019-00563-T

Prof. Hibiki Kawamata
Weill Cornell Medicine
Brain and Mind Research Institute
407 East 61st street, RR508
New York, NY 10065

Dear Dr. Kawamata,

Thank you for transferring your manuscript entitled "Nrf2 signaling links ER oxidative protein folding and calcium homeostasis in health and disease" to Life Science Alliance. The manuscript was assessed by expert reviewers at another journal twice before, and the editors transferred those reports to us with your permission.

The reviewers who evaluated your work at the other journal were concerned by the model used / the link to ALS. We concluded that these concerns are not precluding publication in Life Science Alliance given our transparent process (all reviewer comments will be published alongside the manuscript). We would thus like to invite you to submit a slightly further revised version for publication here, addressing the remaining concerns of the reviewers. Specifically, we would expect:

- that an ANOVA analysis gets performed (rev#1)
- that alternative explanations get discussed (rev#1)
- that the data on neurons get included in the main manuscript (rev#2)
- that the survival rate gets normalized differently (if possible with data at hand) or that potential caveats for this part get discussed

Please include text changes to address the other comments of reviewer #1 and #2.

We would be happy to discuss the individual revision points further with you should this

be helpful.

Thank you for this interesting contribution to Life Science Alliance. We are looking forward to receiving your revised manuscript.

Sincerely,

B. MANUSCRIPT ORGANIZATION AND FORMATTING:

2nd Editorial Decision

18 October 2019

October 18, 2019

RE: Life Science Alliance Manuscript #LSA-2019-00563-TR

Prof. Hibiki Kawamata
Weill Cornell Medicine
Brain and Mind Research Institute
407 East 61st street, RR508
New York, NY 10065

Dear Dr. Kawamata,

Thank you for submitting your revised manuscript entitled "Nrf2 signaling links ER oxidative protein folding and calcium homeostasis in health and disease". I appreciate the introduced changes. I think reviewer #1 would have preferred further softening of the language in regard to disease-relevance, but I accept that this is a matter of opinion. I would thus be happy to publish your paper in Life Science Alliance pending final revisions necessary to meet our formatting guidelines:

- please provide your manuscript file in docx format
- please note that we only have supplementary figures at Life Science Alliance, please change from EV to S nomenclature
- please add a callout in the manuscript text to figure S6G (current fig EV 6G)
- please add a callout in the manuscript text to the table
- note that some of the scale bars in the figures are hard to see, I would like to encourage you to alter these a bit

A. FINAL FILES:

-- Summary blurb (enter in submission system): A short text summarizing in a single sentence the study (max. 200 characters including spaces). This text is used in conjunction with the titles of papers, hence should be informative and complementary

to the title. It should describe the context and significance of the findings for a general readership; it should be written in the present tense and refer to the work in the third person. Author names should not be mentioned.

B. MANUSCRIPT ORGANIZATION AND FORMATTING:

Sincerely,

3rd Editorial Decision

21 October 2019

October 21, 2019

RE: Life Science Alliance Manuscript #LSA-2019-00563-TRR

Prof. Hibiki Kawamata
Weill Cornell Medicine
Brain and Mind Research Institute
407 East 61st street, RR508
New York, NY 10065

Dear Dr. Kawamata,

Thank you for submitting your Research Article entitled "Nrf2 signaling links ER oxidative protein folding and calcium homeostasis in health and disease". It is a pleasure to let you know that your manuscript is now accepted for publication in Life Science Alliance. Congratulations on this interesting work.

DISTRIBUTION OF MATERIALS:

Again, congratulations on a very nice paper. I hope you found the review process to be constructive and are pleased with how the manuscript was handled editorially. We look forward to future exciting submissions from your lab.

Sincerely,
